# Metabolic Modeling Identifies a Novel Molecular Type of Glioblastoma Associated with Good Prognosis

**DOI:** 10.3390/metabo13020172

**Published:** 2023-01-24

**Authors:** Qiu Shen, Hua Yang, Qing-Peng Kong, Gong-Hua Li, Li Li

**Affiliations:** 1The First Hospital of Kunming, Kunming 650600, China; 2State Key Laboratory of Genetic Resources and Evolution/Key Laboratory of Healthy Aging Research of Yunnan Province, Kunming Institute of Zoology, Chinese Academy of Sciences, Kunming 650223, China; 3The Third People’s Hospital of Yunnan Province, Kunming 650600, China

**Keywords:** metabolic modeling, metabolic pathway, GPMM, glioblastoma, cancer prognosis

## Abstract

Glioblastoma (GBM) is one of the most aggressive forms of cancer. Although IDH1 mutation indicates a good prognosis and a potential target for treatment, most GBMs are IDH1 wild-type. Identifying additional molecular markers would help to generate personalized therapies and improve patient outcomes. Here, we used our recently developed metabolic modeling method (genome-wide precision metabolic modeling, GPMM) to investigate the metabolic profiles of GBM, aiming to identify additional novel molecular markers for this disease. We systematically analyzed the metabolic reaction profiles of 149 GBM samples lacking IDH1 mutation. Forty-eight reactions showing significant association with prognosis were identified. Further analysis indicated that the purine recycling, nucleotide interconversion, and folate metabolism pathways were the most robust modules related to prognosis. Considering the three pathways, we then identified the most significant GBM type for a better prognosis, namely N^+^P^−^. This type presented high nucleotide interconversion (N^+^) and low purine recycling (P^−^). N^+^P^−^-type exhibited a significantly better outcome (log-rank *p* = 4.7 × 10^−7^) than that of N^−^P^+^. GBM patients with the N^+^P^−^-type had a median survival time of 19.6 months and lived 65% longer than other GBM patients. Our results highlighted a novel molecular type of GBM, which showed relatively high frequency (26%) in GBM patients lacking the IDH1 mutation, and therefore exhibits potential in GBM prognostic assessment and personalized therapy.

## 1. Introduction

Glioblastoma (GBM) is the most common and devastating primary brain cancer [1]. This disease has a universally fatal prognosis, with over 85% of patients dying within two years [2]. As such, much work has been carried out to understand GBM better, as well as to develop effective treatments and improve the survival time of GBM patients. However, while many drugs and targets have been proposed, most have failed [3]. Thus, finding effective therapies against this lethal cancer remains a considerable challenge.

GBM patients can be classified into different types based on the mutation profiles of different molecular markers [4]. As an important molecular marker, IDH1 mutation in GBM indicates a better prognosis and a potential target for effective treatment, such as targeted immunotherapy for tumor-specific epitopes of IDH1 mutation [5]. However, the IDH1 mutation, which is associated with younger age, only occurs in 5%–10% of all GBMs [5], with the vast majority being IDH1 wild-type. Thus, identifying new molecular markers for prognosis in IDH1-wild-type GBMs could help generate personalized therapies and improve patient outcomes.

An effective way to identify new molecular markers and generate new treatments for malignancies is by targeting cancer metabolic dysfunction, an emerging cancer hallmark [6]. Metabolic dysfunction data (metabolomics) include metabolic profiles and biochemical mechanisms underlying metabolic reprogramming in cancer [7]. Recently, metabolomics and metabolic fluxes have been found to correlate with the GBM patient outcome [8,9,10]. In the current study, we hypothesized that GBMs with different molecular characteristics may have distinct metabolic profiles and may display variable prognoses. Thus, we focused on GBMs lacking the IDH1 mutation and attempted to identify a new molecular marker independent of the IDH1 mutation by determining their metabolic features. As metabolites and gene expression lack correlation [11], it is difficult to infer metabolic profiles using typical gene expression analysis, such as differential gene expression and pathway analyses. Recently, we have developed a novel metabolic modeling method, termed genome-wide precision metabolic modeling (GPMM), by integrating enzyme kinetics, metabolites, and proteomics, which can infer metabolic profiles using regular gene expression analysis and has been successfully applied to disclose metabolic profiles in centenarians [12].

Here, we used GPMM to perform genome-wide metabolic modeling of GBMs lacking IDH1 mutation and investigated their metabolic profiles. We found a new GBM type independent of the IDH1 mutant, named N^+^P^−^, which exhibited a significantly better prognosis than other GBM patients. This “good” prognosis type could potentially be used to personalize cancer therapy and design anti-GBM drugs.

## 2. Materials and Methods

### 2.1. GBM Gene Expression Data

The GBM gene expression data were downloaded from the Genomic Data Commons (GDC) web server using the R package in The Cancer Genome Atlas (TCGA) biolinks. As the GPMM only supports FPKM input, we only downloaded FPKM values from the GDC. We obtained 174 samples that have the FPKM values. The source code for downloading the expression data can be found at: https://github.com/GonghuaLi/Code_for_publications/tree/master/GPMM_Glioblastoma (accessed on 17 July 2020). The clinical dataset and mutation information were downloaded from the TCGA cbioportal website (https://www.cbioportal.org/study/summary?id=gbm_tcga). Of these 174 samples, 149 lacked IDH1 mutations and remained for further analysis.

### 2.2. Metabolic Modeling

The GPMM software version 1.0 was downloaded from the GitHub website (https://github.com/GonghuaLi/GPMM). The metabolic uptake rates of the brain were obtained from previous literature and are shown in Appendix A. As the parameters in GPMM have been optimized in our previous study, the GPMM parameters were also set as per our centenarian study [12]. All of the settings were shown in “pars.txt” in our upload code directory (https://github.com/GonghuaLi/Code_for_publications/tree/master/GPMM_Glioblastoma). Briefly, the human metabolic map, metabolic flux unit, and gene expression type parameters were set to curated Recon3D [13], µmol/min/L, and FPKM, respectively. The GPMM first used a mathematical model to estimate relative protein abundance [14], then used Michaelis–Menten kinetics to fix the upper and lower flux bounds for each enzyme related to metabolic reactions. Next, the biomass reaction and ATP production were set as the objective function to constrain the fluxes of the metabolic network. Finally, Markov chain Monte Carlo (MCMC) sampling was performed to obtain all metabolic reaction fluxes for each sample. To evaluate the performance of our method, we have previously carried out a benchmarking analysis using experimental measurements from the NCI-60 cancer cell lines and compared GPMM with other existing methods. The result showed that GPMM dramatically improved the performance of the modeling with an R^2^ of 0.86 between the predicted and experimental measurements over the performance of existing methods (for details, see ref. [12]).

### 2.3. Survival Analysis

The prognosis profiles of metabolic fluxes were analyzed using the R package from Bioconductor (https://www.bioconductor.org). The GBM patients were classified using a median flux cutoff, with samples divided into two subtypes: “high flux” (above median) and “low flux” (below median) subtypes. Log-rank survival *p*-values and visualizations were performed using the R survminer package. A log-rank *p*-value < 0.01 considered that the flux is significantly associated with GBM prognosis.

### 2.4. Differential Gene Expression and Metabolic Flux Analysis

We first calculated the log2-transformed fluxes and then used the eBayes method in the Limma R package in Bioconductor for differential flux analysis. The *p*-value cutoff was set as 0.01. By using a similar procedure, differential gene expression analysis was also conducted using the eBayes method in the Limma R package (version 3.44.3) in Bioconductor.

### 2.5. Statistical Analysis for Identifying Metabolic Reactions Related to Prognosis

To obtain the robust metabolic reactions related to prognosis, we filtered the reactions using the following criteria: (1) reaction flux significantly associated with prognosis (*p* < 0.01); (2) reaction not singular, i.e., correlated with more than one significant reaction (R > 0.8). After applying these strict criteria, we obtained 12 metabolic modules.

### 2.6. Flux Pathway Analysis

For flux pathway analysis, we used a similar method to the differential abundance (DA) score to identify the overall flux changes in a pathway [11]. We used a weighted differential abundance (*WDA*) score to analyze pathway abundance in GBM, which was calculated as follows:(1)$$WDA=100\times Weight\times \frac{No.fluxes\;increased-No.\;fluxes\;decreased}{No.\;fluxes\;in\;this\;pathway}$$
where,
(2)$$Weight=\frac{No.\;fluxes\;increased\;in\;this\;pathway+No.\;fluxes\;decreased\;in\;this\;pathway}{No.\;total\;significant\;fluxes\;in\;metabolic\;network}$$

## 3. Results

### 3.1. Metabolic Profiles of GBM by Metabolic Modeling

To investigate which metabolic pathways are associated with the prognosis of GBM, we have analyzed the expression profile of metabolic-related genes of the 149 GBM patients (all lacking IDH1 mutation) based on their transcriptomic data (see Section 2). The metabolism-related genes considered here were from the state-of-the-art human metabolic map, Recon 3D, which contains 3288 genes. Among these genes, only 23 were significantly associated with overall survival time in GBM (Appendix A), which did not display enrichment in any metabolic pathway, suggesting that the signals captured by gene expression data are quite different from that of metabolic modeling. So, additional methods are required to study the metabolic changes in GBM.

We then used our recently developed metabolic modeling method GPMM to estimate the metabolic profiles of GBM patients. The GPMM uses a state-of-the-art human metabolic map and analysis toolbox (Cobra 3.0) [15]. We obtained the core GBM metabolic network containing 3949 reactions and 1871 metabolites. To investigate which metabolic reactions were associated with prognosis, we calculated prognosis-related metabolic reactions to identify new GBM type(s) showing better prognosis.

Results showed that 48 metabolic reactions were linked with overall survival time in GBM (Appendix A), with 25 and 23 metabolic reactions found to be negatively and positively correlated with survival time, respectively. Several metabolic pathways were correlated with poor prognosis, including NAD metabolism, folate metabolism, mitochondria transport, and fructose/mannose metabolism (Figure 1A). Interestingly, some metabolic pathways, including starch and sucrose metabolism, leukotriene metabolism, nucleotide interconversion, and glutathione metabolism, were associated with better outcomes.

### 3.2. Robust Metabolic Modules Related to Prognosis

After applying two strict criteria (see Section 2), we obtained 12 robust metabolic modules related to prognosis. Among them, three modules (purine recycling, nucleotide interconversion, and folate metabolism) all contained at least three significant reactions (Figure 1B). Specifically, the purine recycling metabolic module (including the exchange reaction of 5-O-phosphonato-alpha-D-ribofuranosyl diphosphate: EX_prpp(e), adenine reversible transport: ADEt, and AMP-pyrophosphate phosphoribosyltransferase: r0051) (Figure 1B) and folate metabolism module (including dihydrofolate-NAD+ oxidoreductase: r0512, dihydrofolate-NADP+ oxidoreductase: r0514, and facilitated diffusion of folate: r0962) were negatively correlated with survival time, whereas the nucleotide interconversion module (including cytidylate kinase (CMP, CTP): CYTK6, nuclear cytidylate kinase (CMP): CYTK5n, and UMP kinase (UTP): UMPK3) was positively associated with survival time. The reactions in each module were highly correlated with each other (R > 0.8).

### 3.3. Defining New GBM Type with Better Prognosis Than IDH1 Mutant-Type

The above three modules were all independent of IDH1 mutation, as all GBM samples with this mutation were previously disregarded. To define a new GBM type with a better prognosis, we further performed the survival analysis of reactions in these three modules. Since the fluxes in each module are highly correlated (rho > 0.8, Figure 1B), we first selected a representative reaction for each module for prognosis analysis. For example, Ex_prpp(e), r0512, and CYTY6 represented the purine recycling module, folate metabolism module, and folate metabolism module, respectively. For each reaction, the patients were classified into two types based on the flux values, adopted to evaluate whether the reaction activity is associated with the prognosis. In this process, patients with a flux larger than the median values are defined as having a higher flux class, and vice versa. The results showed that patients with lower Ex_prpp(e) reaction (purine recycling module) or r0512 reaction (folate metabolism module) but higher CYTK6 reaction (nucleotide interconversion module) lived much longer than other patients (*p* = 0.001, 0.0044, and 0.0012, respectively; Figure 2A–C).

To determine whether a combination of markers could further improve the significance of discrimination in GBM prognosis, we combined two modules out of the three modules. For example, in the combination of nucleotide interconversion (representative reaction: CYTK6) and purine recycling (representative reaction: EX_prpp(e)) (Figure 2D), we compared the prognosis between the patients with higher nucleotide interconversion activity (N^+^) but lower purine recycling activity (P^−^) and the patients with higher nucleotide interconversion activity (N^−^) but lower purine recycling activity (P^+^). We found that the combination of nucleotide interconversion and purine recycling (NP) has a hazard ratio (HR) of 0.32 (log-rank *p* = 4.7 × 10^−7^, Figure 2D). For the combination of purine recycling and folate metabolism, we obtained an HR of 2.67(log-rank *p* = 1.8 × 10^−5^, Figure 2E). For the combination between nucleotide interconversion and folate metabolism, the HR is 0.43 with a log-rank *p*-value of 1.0 × 10^−4^ (Figure 2F). This suggested that the combination of nucleotide interconversion and purine recycling exhibited the best prognosis for IDH1 wild-type GBM (Figure 2D–F). This molecular characteristic, termed N^+^P^−^ here, showed higher nucleotide interconversion activity (N^+^) but lower purine recycling activity (P^−^). Patients with N^+^P^−^ lived significantly longer than those with N^−^P^+^ (log-rank *p* = 4.7 × 10^−7^). In addition, N^+^P^−^ patients exhibited a median overall survival time of 19.6 months and lived 116% (10.3 months) longer than N^−^P^+^ patients (Figure 2D), 34% (12.9 months) longer than P^+^ patients, and 65% longer than other GBM patients.

### 3.4. Metabolic Profiles of N^+^P^−^ Type

To examine the metabolic profiles of N^+^P^−^, we compared the differences between N^+^P^−^ and N^−^P^+^ and identified 151 up-regulated fluxes and 154 down-regulated fluxes (Figure 3A). Among them, the most significantly up-regulated reactions were CYTK6 and CYTK5n, both belonging to the nucleotide interconversion module. However, the most significantly down-regulated reactions were EX_prpp(e), ADEt, and r0051, all belonging to the purine recycling metabolic module.

By obtaining a genome-wide metabolic change map (Figure 3B,C), we found that the elevated metabolic reactions were related to the synthesis and metabolism of androgen and estrogen, hyaluronan metabolism, lysosomal transport, nucleotide interconversion, and tyrosine metabolism (Figure 3B). Furthermore, the decreased metabolic reactions were related to the metabolisms of inositol phosphate, phosphatidylinositol phosphate, pyrimidine, glutathione, and nuclear transport (Figure 3B).

### 3.5. Gene Expression Features of N^+^P^−^ Type

We performed differential gene expression analysis between N^+^P^−^ and N^−^P^+^ to guide GBM treatment through gene expression regulation potentially. Results identified 1795 up-regulated genes and 1797 down-regulated genes in N^+^P^−^ compared with N^−^P^+^ (Figure 4A). Enrichment analysis showed that the up-regulated genes were enriched in carbon metabolism, oxidative phosphorylation, glutathione metabolism, and the TCA cycle (Figure 4B). In addition, the AMPK signaling, phospholipase D signaling, endocrine resistance, and phospholipid metabolic process pathways were all down-regulated in N^+^P^−^ (Figure 4C). Among the up-regulated KEGG pathways, carbon metabolism was the most significant. Carbon metabolism includes one-carbon metabolism and central carbon metabolism, which are both related to cancer cell survival [16,17]. Of note is that AMPK was the most notable down-regulated KEGG pathway. AMPK functions as a “conditional” tumor suppressor and “contextual” tumor promoter and exerts a double role in the development/progression of cancer cells by activating different downstream pathways in a context-specific manner [18].

## 4. Discussion

Cellular metabolism is a complex network containing thousands of reactions and thousands of genes [19,20]. Systematically characterizing the entire metabolic network by using routine gene expression analysis remains challenging [21], largely due to the poor correlation between cellular metabolite abundance and gene expression [11]. Meanwhile, it was established that metabolic dysfunction plays an important role in the onset, development, and metastasis of GBM [10]. Systematically understanding the metabolic characteristics of GBM would help to provide deeper insights into the underlying mechanism or identify novel molecular markers of the disease. In fact, although the well-known IDH1 mutation is a good molecular predictor for favorable outcomes of GBM [4,10], unfortunately, the majority of GBM patients lack the IDH1 mutation. Therefore, discovering additional novel molecular markers could greatly help the prognostic assessment and individual treatment of most GBM patients.

Here, we systematically estimated and analyzed metabolic reactions associated with prognosis in IDH1-wild-type GBMs using our newly developed GPMM approach [12]. Based on the most recently updated Recon 3 [13], our method can capture thousands of reactions and facilitate a better understanding of the metabolic profiles of certain diseases of interest. We then estimated and analyzed the metabolic reaction profiles of 149 GBM samples lacking the IDH1 mutation. Among the 3949 reactions identified, 48 showed significant association with prognosis. Further analysis of these 48 reactions showed that the purine recycling, nucleotide interconversion, and folate metabolism pathways were the most robust modules related to prognosis. We then obtained the most significant molecular characteristics associated with GBM prognosis and defined a new molecular type (N^+^P^−^), which exhibits higher nucleotide interconversion (N^+^) and lower purine recycling (P^−^). Given that our metabolic modeling was constrained by the objective function of biomass reaction and ATP production using flux balance analysis (FBA), it is possible that the identified reaction modules might be related to biomass reaction and ATP production. We performed a large-scale Markov chain Monte Carlo (MCMC) sampling after the FBA analysis to address this issue. This strategy enables the GPMM to be more robust and, thus, obtains more reactions that are less related to the biomass equation and ATP production itself. For example, we used typical FBA analysis to find that 258 reactions are essential for biomass reaction and ATP production (Appendix A). However, after large-scale MCMC sampling was performed, we obtained 3949 reactions for the next step of the analysis [12,13] (Appendix A), suggesting that the influence of FBA on the obtained results is, to some extent, adjusted. In fact, the highlighted reactions in this study, such as nucleotide interconversion and purine recycling, are less related to biomass reaction and ATP production.

Interestingly, the molecular characteristics of the newly defined GBM type, namely, purine recycling and nucleotide interconversion, have already been documented to play important roles in carcinogenesis, with either lower purine recycling (P^−^) or higher nucleotide interconversion (N^+^) showing anticancer effects. Specifically, purine is an essential substrate for nucleotide synthesis [22,23] and provides the necessary energy and cofactors to promote cell survival and proliferation [8,22]. Purine can be produced by the complementary salvage and de novo biosynthetic pathways. The complementary salvage pathway accounts for most of the cellular requirements for purine by recycling degraded bases [21]. Overactivation of purine recycling can generate additional inosine monophosphate (IMP), which contributes to the production of various intermediates, such as adenosine, AMP, GMP, and inosine [21]. Adenosine is known to participate in tumorigenesis [24], whereas AMP and GMP are important second messengers in cellular signal transduction systems. Thus, these intermediates could contribute to carcinogenesis. Supporting evidence comes from the observation that patients with higher purine recycling activation have a significantly poorer prognosis (Figure 2A).

We also investigated the protein abundance of nucleotide interconversion-associated genes (HPRT1) and purine recycling-associated genes (CMPK1) and analyzed the relationship between the abundance of these proteins and GBM prognosis by using the GBM proteome data reported recently [25]. Unfortunately, the protein abundances of these genes were not associated with GBM patient survival (*p* > 0.05), likely because metabolic characteristics are the main factor in our definition of GBM subtypes. A more comprehensive investigation of metabolism and fluxes would be indispensable to address this issue.

As to nucleotide interconversion, by which extracellular ATP is produced from ADP, AMP, GTP, and UTP [26], extracellular ATP is considered an anticancer substance [27]. It can inhibit the growth of a variety of human tumors, induce resistance of nonmalignant tissues to chemotherapy and radiation therapy [28], and may direct chemotherapeutic drugs toward brain tumor cells [29]. Higher nucleotide interconversion activity produces more extracellular ATP and may lead to better outcomes due to the anticancer effects of extracellular ATP. Indeed, our results showed that patients with higher nucleotide interconversion activity lived significantly longer (log-rank *p* = 0.0012). To the best of our knowledge, this is the first study to link nucleotide interconversion with cancer prognosis.

To obtain the gene expression profile of the N^+^P^−^ type, we performed gene ontology (GO) enrichment analysis. The KEGG pathway analyses of GBM samples belonging to the N^+^P^−^ type revealed that the gene expression profiles were significantly enriched in the AMPK signaling pathway. Recent GBM research demonstrated that AMPK rewires GBM stem cell metabolism and promotes tumor growth [30], supporting our findings that the AMPK signaling pathway was down-regulated in N^+^P^−^.

Although we have identified a novel GBM subtype using the GPMM approach, using metabolic reaction fluxes as clinical biomarkers is still challenging, partly due to technical limitations in measuring them robustly and at scale. Therefore, finding other easily measurable value(s) closely related to flux biomarkers, such as metabolites, protein abundance, etc., collecting more high-quality data, and further experimental validation will be our next step in the future.

## 5. Conclusions

Using our newly developed GPMM approach, we investigated the metabolic profiles of GBM patients lacking IDH1 mutation and successfully identified 48 metabolic reactions significantly associated with prognosis. Importantly, we defined a novel GBM type (N^+^P^−^) independent of the IDH1 mutant type, exhibiting a significantly better prognosis than other GBM patients. This type displayed a relatively high frequency (26%) in GBM patients lacking the IDH1 mutation, indicative of its considerable potential in GBM prognostic assessment. The characteristics of the N^+^P^−^ type suggest the possibility of personalized therapies for IDH1 wild-type GBM by improving nucleotide interconversion and inhibiting purine recycling.

## Figures and Tables

**Figure 1 metabolites-13-00172-f001:**
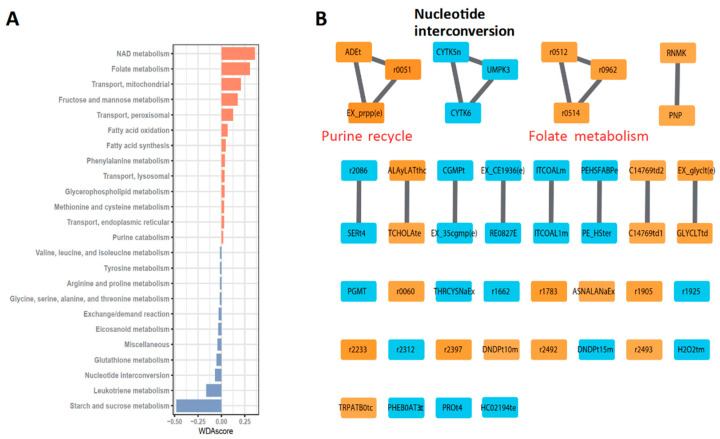
Prognosis analysis of genome−wide precision metabolic modeling in GBM. (**A**): Metabolic pathways correlated with GBM prognosis. Pathways negatively and positively correlated with survival time are colored orange and blue, respectively. (**B**): Metabolic modules associated with GBM prognosis. Modules negatively and positively associated with GBM patient survival time are also colored orange and blue, respectively.

**Figure 2 metabolites-13-00172-f002:**
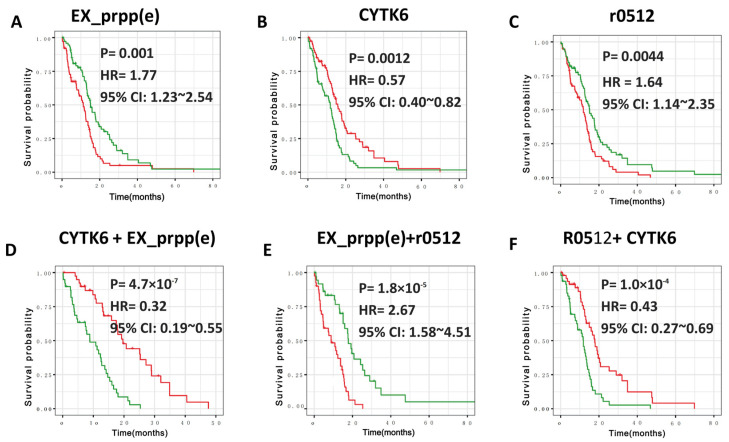
Survival profiles of three different reactions in GBM. (**A**–**C**): Survival curves of EX_prpp(e), CYTK6 and r0512. Patients with high and low flux are colored red and green, respectively. (**D**): Survival curves of combining CYTK6 with EX_prpp(e) reactions, patients with high and low flux are colored red and green, respectively. (**E**): Survival curves of combining EX_prpp(e) with r0512 reactions, patients with high and low flux are colored red and green, respectively. (**F**): Survival curves of combining R0521 with CYTK6, patients with high and low flux are colored red and green, respectively.

**Figure 3 metabolites-13-00172-f003:**
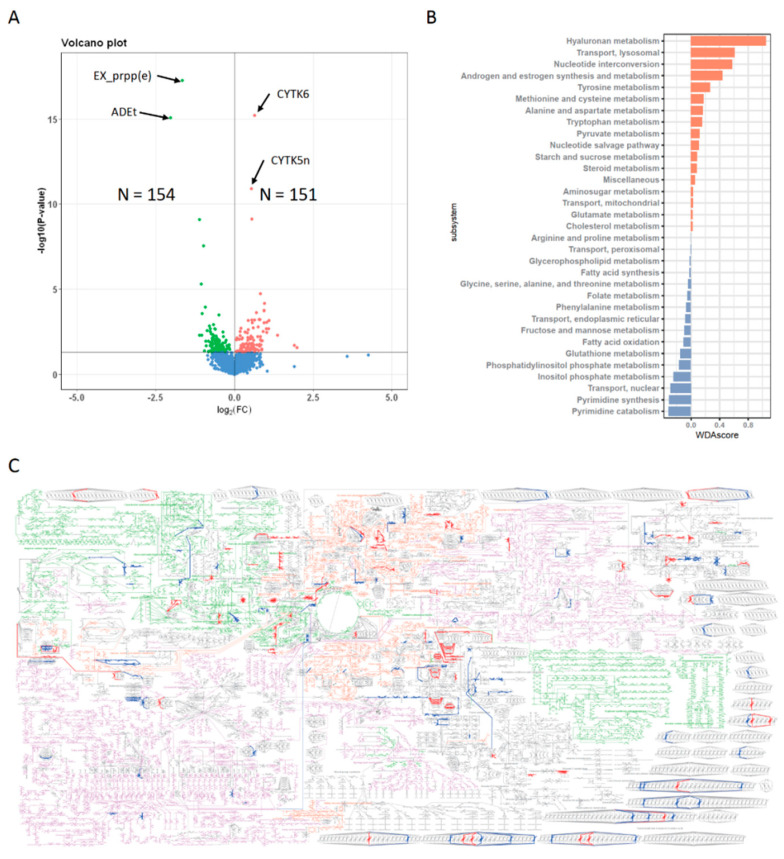
Metabolic profiles of N^+^P^−^. (**A**): Volcano plot of differential reactions between the N^+^P^−^ and N^−^P^+^. Up- and down-regulated reactions in N^+^P^−^ are colored red and green, respectively. (**B**): Differential metabolic pathways between the N^+^P^−^ and N^−^P^+^. (**C**): Overall metabolic changes in N^+^P^−^. The overall metabolic map of Recon was downloaded from the Virtual Metabolic Human Database (https://vmh.uni.lu/). Three general metabolic pathways, including carbohydrate metabolism, amino acid metabolism, and fatty acid metabolism, are colored green, orange, and purple, respectively. The significantly up- and down-regulated reactions in N^+^P^−^ are colored bold red and bold blue, respectively.

**Figure 4 metabolites-13-00172-f004:**
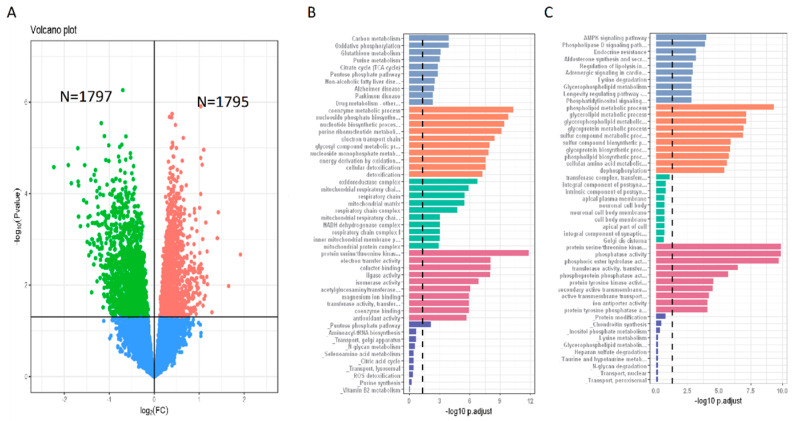
Gene expression profile of N^+^P^−^ GBM samples. (**A**): Volcano plot of differential gene expression between the N^+^P^−^ and N^−^P^+^. (**B**): GO enrichment analysis of up-regulated genes (1795 genes). (**C**): GO enrichment analysis of down-regulated genes (1797 genes).

## Data Availability

The datasets generated and/or analyzed during the current study are available on the Genomic Data Commons (GDC) web (https://portal.gdc.cancer.gov/) and the TCGA cbioportal website (https://www.cbioportal.org/study/summary?id=gbm_tcga, accessed on 17 July 2020). The source code in this study is available at GitHub (https://github.com/GonghuaLi/Code_for_publications/tree/master/GPMM_Glioblastoma). Further information and requests for resources and software should be directed to and will be fulfilled by Gong-Hua Li (ligonghua@mail.kiz.ac.cn).

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
