# Peer review of "Metabolic Modeling Identifies a Novel Molecular Type of Glioblastoma Associated with Good Prognosis"

_metabolites, 2023, doi:10.3390/metabo13020172_

Round 1

Reviewer 1 Report

Authors introduce a possible subclass of GBM in regard to its metabolic profile.

The introduction includes IDH1 as important molecular marker and could integrate the common classification scheme as to WHO 2021 (e.g. https://doi.org/10.1093/neuonc/noab106) introducing several subclasses based on the genetic profile.

Additional examples of metabolic profiles for diseases distinct from brain cancer could be added as background information.

Possible additions to references in intro to underline method: Xu, X. et al. and Schroeder et al. already promote metabolomics and metabolic flux analysis (s.a. 2021 https://doi.org/10.1038/s41419-021-03543-9, 2020 https://doi.org/10.1117%2F1.JBO.25.3.036502, 2022 https://doi.org/10.1093/neuonc/noac042)

Minor spell check recommended s.a. "showed significantly association" => "significantly showed association", missing space between "Modeling(GPMM)"

Figure 3 could be integrated as pdf or other vector-based format in order to be scalable and readable!

Individual reactions p4 144+ could be introduced and described in detail.

p7 253 needs reference(s).

Reviewer 2 Report

The authors used a metabolic modeling method  (Genome-wide Precision Metabolic Modeling, GPMM) to investigate the metabolic profiles of 149 Glioblastoma samples lacking IDH1 mutation. Three metabolic pathways were related to prognosis. The topic is of interest but, in my opinion, requires experimental validation. Manuscripts consisting solely of bioinformatics can not be accepted.

Reviewer 3 Report

The enclosed manuscript aims to suggest a novel prognostic metabolic combination for GBM with wild-type IDH1 expression, in which a nucleotide interconversion-high plus purine recycling-low profile may lead to a worse prognostic outcome. This study was initiated with a genome-wide bioinformatics analysis and landed on the metabolomics conclusion, suggesting several metabolic events according to the genotype. Retrospectively, using such criteria can effectively indicate the prognosis of GBM patients, according to the authors' report. This can be an important finding for clinical impacts, but some concerns are listed as follows:

1) The entire study was based on one-off sequencing data. Whether or not this study can be representative remains a question mark. The authors may consider obtaining more cases for a solid suggestion. 

2) This study used only sequencing results and sophisticated bioinformatics approaches to name the metabolic events. However, whether such genes bring certain metabolic functions remains unclear. It is not as convincing as what it looks like by using the name of "Metabolic Models." Instead, it might be more precise to indicate the genes involved, such as CYTK6. More discussion can be made according to the concerns. 

3) According to the figure 2, it is apparent that CYTK6 dominates the prognostic evaluation where higher CYTK6 results in better outcomes, regardless of combination with any other markers. Therefore, what have inspired the authors to include other markers? Meanwhile, why the inclusion of other markers in an opposite trend can further expand the CYTK6-led prognostic analysis? It is a bit confusing in both the description and the plots for reading. 

4) Although the bioinformatics results are straightforward, it is suggested to provide more correlated biomarkers, such as metabolic indicators or clinical samples. 

Reviewer 4 Report

In this paper, the authors, through bioinformatics analysis, proposed new possible therapeutic targets from cellular metabolism in glioblastoma.

Main concerns:

It is unclear what samples and datasets were used to perform the analysis, in fact:

- the materials and methods lack the amount and references of the downloaded datasets, how they were selected (what are the characteristics of the patients?), the characteristics of the datasets (are they WGS? RNA Seq? where are the references of the transcriptomics data?). Also, how many patients were analyzed in total? The table with the clinical data of the patients analyzed is missing. Also missing is information on how the analyses between metabolic reactions and patient survival were done, how were they analyzed? 

- It is unclear which datasets were selected because line 115 states that only samples without mutation for IDH1 were selected, but in line 128 the authors write that they removed samples mutated for IDH1 from the analysis. Therefore, the results reported in Sections 3.2 and 3.3 are not clear.

- In addition, the authors should explain why they chose to use the GPMM model setting used in their previous work and should report the settings in the materials and methods.

- The section on statistical analysis is completely missing.

- In Figure 3A, the number of patients does not match what is written in the text. The data presented in lines 232 and 233 also do not match what is written in the results. In Figure 2D, the number of patients is not shown between the graphs; the authors should explain why.

- The discussion should be completely revised as it is an additional description of the results obtained with explanations that should be present in the materials and methods or in the results part. It is recommended that the discussion be rewritten with the above comments in mind.

- The supplementary data should be presented as tables and not as excel files.

Minor issues:

- Figures 3 and 4 need to be improved in terms of quality. Also, Figure 2 does not have consistent formatting.

- There is an error on line 56

- Lines 174 and 176: are the names correct?

Round 2

Reviewer 2 Report

Further topics should be at least discussed. i.e. the role of AltProt in the regulation of pathological function is growing (Nucleic Acids Res2020 Aug 20;48(14):7864-7882; Nat Commun

2022 Nov 4;13(1):6665.

 doi: 10.1038/s41467-022-34208-6). 

Reviewer 3 Report

The authors have addressed my comments properly. Thank you.

Reviewer 4 Report

The work seems to have improved after the review, but some concerns are still present.

- The "Statistical Analysis" section needs to be in the materials and methods section;
- Section 3.3 and Figure 2 are still unclear, and the text should be improved by better explaining what was analyzed. Also, it would be advisable to remove the number of regulated genes (2A-C) and patients (D-F) from the picture and insert the hazard ratio and CI.
